# Current Understanding of Protein Aggregation in Neurodegenerative Diseases

**DOI:** 10.3390/ijms262110568

**Published:** 2025-10-30

**Authors:** Chen Hu, Menghan Lin, Chuangui Wang, Shengping Zhang

**Affiliations:** Biomedical Research Institute, School of Life Sciences and Medicine, Shandong University of Technology, Zibo 255049, China; chenhu2023@sdut.edu.cn (C.H.); 23510031038@stumail.sdut.edu.cn (M.L.); cgwang24@sdut.edu.cn (C.W.)

**Keywords:** protein aggregates, neurodegenerative diseases, oligomers, aggregate clearance, therapeutic strategies

## Abstract

Protein aggregates are central to the pathogenesis of neurodegenerative diseases such as Alzheimer’s disease and Parkinson’s disease. This comprehensive review explores the mechanisms of protein misfolding and aggregation, their prion-like propagation, and the critical role of oligomeric species in neurotoxicity. It further examines cellular clearance pathways, including the ubiquitin–proteasome system and autophagy, alongside the regulatory functions of molecular chaperones. The review also covers advanced diagnostic imaging and biomarker techniques, as well as emerging therapeutic strategies such as pharmacological agents, gene therapy, and immunotherapy. Controversies regarding the toxicity of aggregates and future directions, including novel degradation technologies and targeted therapeutic approaches, are discussed. By integrating current knowledge, this review aims to provide a broad yet detailed overview of the field, highlighting both established concepts and promising avenues for research and treatment.

## 1. Introduction

Protein aggregation is a defining pathological feature of numerous neurodegenerative diseases, including Alzheimer’s disease, Parkinson’s disease, amyotrophic lateral sclerosis, Huntington’s Disease and others [1,2,3,4,5,6,7,8]. These aggregates, composed of misfolded proteins such as amyloid-β, tau, and α-synuclein, disrupt cellular homeostasis, impair neuronal function, and ultimately contribute to cell death [9,10,11,12,13,14]. Beyond their structural roles, protein aggregates exhibit prion-like properties, facilitating their propagation across neural networks and exacerbating disease progression [15,16,17,18,19,20,21,22]. While historically considered inert end-products, emerging evidence underscores the heightened toxicity of soluble oligomeric species, which are now regarded as primary drivers of neurotoxicity [23,24,25,26]. The mechanisms underlying protein aggregation are influenced by genetic predispositions, post-translational modifications, oxidative stress, and impaired protein quality control systems, including the ubiquitin–proteasome system and autophagy. This review comprehensively examines the role of protein aggregates in neurodegenerative pathologies, exploring their formation, clearance, diagnostic detection, and therapeutic targeting, while also addressing ongoing controversies and future research directions aimed at mitigating their devastating effects.

## 2. Perspectives on Protein Aggregation

The study of protein aggregation has been a focal point in understanding the pathogenesis of various neurodegenerative diseases. Perspectives on protein aggregation have highlighted the role of misfolded proteins in disease progression, particularly in conditions such as Alzheimer’s disease (AD), Parkinson’s disease (PD), and amyotrophic lateral sclerosis (ALS) (Table 1).

### 2.1. Oligomers: Structure and Toxicity

Early research indicated that the most basic aggregates consist of a limited number of misfolded protein units, commonly referred to as oligomers. These oligomers are characterized by their amorphous structures, which are rich in exposed hydrophobic regions, rendering them highly reactive and potentially the most hazardous type of aggregate [9,26,32]. Oligomers are often metastable and may progress along a misfolding pathway toward the formation of higher-order structures [33]. These advanced structures are composed of numerous protomers and exhibit increased compactness, with hydrophobic residues more densely packed, resulting in exceptionally stable configurations that can persist for extended periods, even decades, within biological systems [34]. The transition from oligomers to fibrils involves multiple intermediates, each marked by a progressive increase in compaction, also described as rigidity. Oligomeric species are in a highly dynamic state, balancing with monomers and fibrils. Many rigid protein aggregates adopt an amyloid structure, characterized by an abundance of β-sheet folds. The β-sheet configuration is a critical structural motif in amyloid fibrils, providing the stability and rigidity necessary for fibril formation. This structural transformation is often initiated by the formation of β-strand seed structures, which act as nucleation sites for further aggregation. In environments with limited solvent, such as those mimicked by reverse micelles, monomeric amyloid-β (Aβ) proteins can form extended β-strands, suggesting a plausible mechanism for amyloid fibril nucleation in the brain [35]. The role of β-sheets in amyloid fibril formation is not limited to structural stability but also extends to their involvement in the pathogenicity of amyloid diseases. The formation of β-sheet-rich oligomers is often associated with increased cytotoxicity, as seen in the neurotoxic amyloid intermediates of Alzheimer’s amyloid-β [36]. Furthermore, certain oligomers serve as intermediates in the formation of amyloid fibrils, whereas others may be terminal products that do not follow the pathway, and some of these could be highly toxic (Figure 1). The distinction between on-pathway and off-pathway oligomers is further highlighted by studies on α-synuclein and amyloid-β (Aβ) peptides. In the case of α-synuclein, certain oligomers inhibit fibril formation by blocking secondary nucleation sites, demonstrating their role as off-pathway species [37]. Similarly, Aβ oligomers can undergo liquid–liquid phase separation, which modulates fibril formation, indicating their involvement in complex aggregation pathways [38]. These findings underscore the diverse roles oligomers can play, either facilitating or hindering fibril formation. The role of these higher-order aggregates in disease pathology remains a subject of ongoing investigation [39].

### 2.2. Prion-like Propagation of Protein Aggregates

These protein aggregates are not only a result of misfolding but also exhibit prion-like properties, enabling them to propagate between cells and exacerbate disease progression [15,17,40,41,42,43]. As previously reviewed, the concept of protein aggregates exhibiting prion-like properties has gained significant attention in recent years, particularly in the context of neurodegenerative diseases. Prions, traditionally associated with transmissible spongiform encephalopathies, are unique infectious agents composed solely of misfolded proteins that propagate by inducing misfolding in normal proteins. This prion-like behavior is not limited to classical prion diseases but extends to other protein misfolding disorders, suggesting a broader applicability of the prion concept [44]. Recent studies have demonstrated that misfolded protein aggregates can self-propagate through a seeding mechanism, akin to prions, in various neurodegenerative diseases such as Alzheimer’s and Parkinson’s diseases. Current research suggests the hypothesis that prions do not induce the misfolding of proteins but rather shift the equilibrium between folding intermediates towards aggregation, supported by several studies that explore the dynamics of protein folding and aggregation. One key study highlights the role of transient folding intermediates in amyloidogenicity, suggesting that these intermediates are crucial in determining whether a protein will aggregate into amyloid fibrils. The study demonstrates that subtle changes in the population of these intermediates can significantly impact the aggregation pathway, thereby supporting the idea that prions shift the equilibrium rather than directly inducing misfolding [45]. These aggregates, including amyloid-β, tau and alpha-synuclein, can induce a self-perpetuating process that leads to the amplification and spreading of pathological protein assemblies at multiple levels [15,17,42,46] (Figure 1). At the molecular scale, the template-induced transformation of a natively folded protein by a polymeric misfolded protein results in the autocatalytic expansion of protein aggregates (blue color). At the cellular scale, the pathology propagates from one cell to another via the transfer of misfolded protein aggregates (blue stars) between neighboring cells, resulting in the regional dissemination of abnormalities. At the organ scale, the progressive spread of pathology (blue stars) among cells culminates in tissue damage, which can extend to remote or distant brain regions through mechanisms such as cell-to-cell contact or the movement of biological fluids, including interstitial fluid, cerebrospinal fluid, or blood.

The prion-like behavior of protein aggregates is further supported by evidence from studies on amyotrophic lateral sclerosis (ALS), where misfolded superoxide dismutase-1 (SOD1) aggregates have been shown to propagate in a prion-like manner. These aggregates can spread between cells and induce misfolding of native SOD1, highlighting the role of prion-like mechanisms in the progression of ALS [21]. Moreover, this prion-like behavior is evident in both intracellular and extracellular environments, contributing to the progression of diseases such as Alzheimer’s, Parkinson’s, and amyotrophic lateral sclerosis (ALS) [21,47].

### 2.3. Cell-to-Cell Transmission and Extracellular Vesicles

The study of cell-to-cell, vesicular uptake and propagation in neurodegenerative diseases has garnered significant attention due to its implications in understanding disease mechanisms and developing therapeutic strategies. The cell-to-cell transmission of pathogenic protein aggregates is a critical mechanism in the progression of neurodegenerative diseases. This process involves the release of misfolded proteins from donor cells, their uptake by recipient cells, and subsequent seeding of new aggregates [48,49]. The molecular heterogeneity of these aggregates, the co-occurrence of different aggregates in some diseases, often referred to as strains, further complicates the pathology, as different strains can lead to distinct clinical manifestations and disease courses [50,51] (Figure 1). This heterogeneity is reminiscent of prion diseases, where different conformations of the same protein can result in varied phenotypic outcomes [52,53]. As previously reviewed, extracellular vesicles (EVs), including exosomes and micro vesicles, are released by various cell types in the central nervous system (CNS) and carry a diverse array of bioactive molecules, such as proteins, RNAs, and lipids, which can influence recipient cells’ function and contribute to disease pathogenesis [54]. As previously examined, microglia-derived EVs have been implicated in the propagation of neurodegenerative pathology, as they can transport misfolded proteins such as α-synuclein and tau, which are associated with Parkinson’s and Alzheimer’s diseases, respectively [55]. However, the role of aggregated proteins and their internalization into vesicles in the propagation of pathology remains a subject of debate. Some studies suggest that the uptake of aggregated proteins into vesicles does not necessarily contribute to the propagation of pathology, challenging the prion-like spread hypothesis. One study that supports this view is an investigation of exosome-associated pathological alpha-synuclein (α-syn) in Parkinson’s disease. It was found that intrastriatal administration of exosome-associated pathological α-syn was not sufficient by itself to cause pathology transmission in wild-type mice. This suggests that exosomes may neutralize the effect of toxic α-syn species, raising questions about their role in disease transmission [56].

## 3. Mechanisms Underlying Protein Aggregation in Neurodegenerative Diseases

The mechanisms of protein aggregation in neurodegenerative diseases are multifaceted and involve a complex interplay of molecular processes that contribute to the pathogenesis of these disorders (Figure 2).

### 3.1. Role of the Cellular Environment

The role of the cellular environment in modulating protein aggregation is also significant. Factors such as oxidative stress, post-translational modifications, and the presence of molecular chaperones can influence the aggregation process (Figure 2). As previously reviewed, oxidative stress and protein aggregation are intricately linked processes that play a significant role in the pathogenesis of various neurodegenerative diseases. The accumulation of reactive oxygen species (ROS) can lead to oxidative modifications of proteins, which in turn can result in protein misfolding and aggregation. This relationship is particularly evident in diseases such as Alzheimer’s and Parkinson’s, where oxidative stress is both a cause and a consequence of protein aggregation [57]. The interplay between these processes creates a vicious cycle that exacerbates cellular dysfunction and contributes to disease progression. One of the key mechanisms by which oxidative stress promotes protein aggregation is through the modification of specific amino acid residues, such as cysteine. The oxidation of cysteine residues can lead to the formation of disulfide bonds, which can destabilize protein structures and promote aggregation. For instance, the oxidation of a single cysteine residue in a globular protein can significantly alter its folding energy landscape, leading to amyloid formation under physiological conditions [58]. Mitochondrial dysfunction is another critical factor that links oxidative stress to protein aggregation. Mitochondria are both a source and a target of oxidative damage, and their impairment can lead to increased ROS production and subsequent protein aggregation. In Parkinson’s disease, for example, mitochondrial oxidative stress has been shown to exacerbate α-synuclein aggregation and spreading, highlighting the role of mitochondria in the pathogenesis of neurodegenerative diseases [59]. Overall, oxidative stress and protein aggregation are closely interconnected processes that contribute to the pathogenesis of neurodegenerative diseases.

### 3.2. Post-Translational Modifications (PTMs)

Post-translational modifications (PTMs) (such as phosphorylation, ubiquitination, acetylation, glycation, sumoylation, oxidation, arginylation, methylation, etc.) are critical biochemical processes that regulate protein function, stability, and interactions, significantly influencing cellular processes and disease pathogenesis. As previously studied, in the context of neurodegenerative diseases, PTMs play a pivotal role in protein aggregation [60,61], a hallmark of these disorders. For instance, α-Synuclein, a protein implicated in Parkinson’s disease, undergoes various PTMs such as phosphorylation, ubiquitination, and acetylation, which modulate its aggregation propensity and pathogenicity. Phosphorylation, particularly at serine 129, is a predominant modification observed in α-synuclein within Lewy bodies, which are pathological hallmarks of PD. This modification is thought to enhance the protein’s aggregation and toxicity, contributing to neurodegeneration [62], while phosphorylation at serine 87 inhibits the aggregation of human α-synuclein and protects against its toxicity [63]. Furthermore, phosphorylation at tyrosine 39 has been shown to affect fibril formation kinetics and alter fibril morphology, indicating its role in modulating α-synuclein aggregation [64]. Ubiquitination is another critical PTM that regulates α-synuclein’s cellular dynamics, influencing its degradation, aggregation, and associated neurotoxicity. Ubiquitinated α-synuclein is targeted for degradation via the proteasomal or lysosomal pathways, and dysregulation of this process can lead to the accumulation and aggregation of the protein, contributing to the formation of Lewy bodies [65,66]. Acetylation, particularly N-terminal acetylation, has been shown to slow down α-synuclein’s aggregation process and alter the morphology of the resulting aggregates. This modification reduces the rate of lipid-induced aggregation and affects the structural properties of fibrillar aggregates, thereby modulating the protein’s aggregation behavior [67,68]. Additionally, acetylation has been implicated in the modulation of α-synuclein’s interactions with lipid membranes, further influencing its aggregation propensity [69]. The interplay between different PTMs, known as PTM crosstalk, adds another layer of complexity to α-synuclein’s regulation. For instance, SUMOylation can counteract ubiquitination, affecting α-synuclein’s degradation and aggregation. This crosstalk between PTMs highlights the intricate regulatory mechanisms that govern α-synuclein’s function and aggregation, offering potential therapeutic targets for modulating its pathogenicity in synucleinopathies [70]. The phosphorylation of Amyloid-β (Aβ) peptides plays a crucial role in modulating the molecular stability of protein aggregates, which in turn contributes to the progression of Alzheimer’s disease (AD). Phosphorylation at specific sites, such as serine 8, has been shown to increase the stability of Aβ aggregates, thereby enhancing their resistance to dissociation and promoting their pathogenic spread in the brain [71]. The aggregation of tau protein in Alzheimer’s disease is closely linked to its hyperphosphorylation, with specific phosphorylation sites correlating with tau multimerization in early disease stages. This underscores the importance of PTMs in the early pathogenesis of neurodegenerative diseases [72]. PTMs are also crucial for regulating RNA-binding proteins like FUS and TDP-43, linked to neurodegenerative diseases such as ALS and FTLD. These modifications affect the proteins’ localization, stability, and interactions, influencing their function and disease involvement. For example, SUMOylation, a PTM, regulates TDP-43’s splicing activity and nucleocytoplasmic distribution, impacting its subcellular localization and recruitment to stress granules, which are vital during cellular stress and neurodegeneration [73]. Phosphorylation affects RNA-binding proteins by modulating interactions in their intrinsically disordered regions (IDRs), influencing phase separation and biomolecular condensate formation. This is crucial for proteins like TDP-43 and FUS, whose aggregation is linked to ALS and FTLD. Phosphorylation alters their RNA-binding properties, impacting RNA metabolism regulation [74]. Additionally, ubiquitination, another PTM, regulates TDP-43 aggregation by affecting its oligomerization, highlighting how PTMs can influence pathological protein aggregation [75]. Moreover, aspartic acid isomerization is a significant post-translational modification that has been implicated in the pathogenesis of neurodegenerative diseases, particularly Alzheimer’s disease (AD). This process involves the conversion of aspartic acid (Asp) to isoaspartic acid (isoAsp), which can alter protein structure and function, potentially leading to protein misfolding and aggregation. The isomerization of Asp residues in tau is a spontaneous chemical modification that can interfere with protein turnover and has been associated with cognitive decline in AD [76]. The isomerization of aspartic acid residues in Aβ peptides has been shown to prevent their degradation by lysosomal proteases, such as cathepsin L. This failure in proteolysis can result in the accumulation of undigested peptide fragments, contributing to lysosomal storage disorders and the progression of AD pathology [77]. The regulation of protein stability by PTMs is another crucial aspect, as these modifications can either promote degradation or stabilize proteins. Post-translational modifications (PTMs) can occur on specific amino acids situated within the regulatory domains of target proteins, thereby influencing the stability of these proteins. These regions, known as degrons, are frequently regulated by PTMs, which function as signals to either accelerate protein degradation (PTM-activated degrons) or inhibit degradation, thereby stabilizing the protein (PTM-inactivated degrons) [78]. For instance, the SUMOylation of tau at lysine 340 as a PTM-inactivated degron, by modulating its phosphorylation and ubiquitination, has been demonstrated to be mutually reinforcing, thereby contributing to the proteolytic stabilization of tau [79]. Understanding the mechanisms of PTM-regulated degrons can enhance the identification of novel drug targets, particularly in diseases where protein aggregation is a central feature [78]. Additionally, in the realm of neurodegenerative diseases, PTMs are not only involved in protein aggregation but also in the regulation of signaling pathways. For example, PTMs regulate the PI3K/Akt/GSK3β and MAPK cascades, which are implicated in the pathogenesis of these disorders. Targeting PTMs with small molecules can reverse misfolded protein accumulation, offering neuroprotective benefits [80,81]. In conclusion, PTMs are integral to the regulation of protein function and aggregation, with significant implications for disease pathogenesis and therapeutic intervention. The complexity and diversity of PTMs necessitate a comprehensive understanding of their roles in cellular processes and disease mechanisms, paving the way for novel therapeutic strategies targeting these modifications.

### 3.3. Genetic Factors

Genetic factors also play a significant role in the aggregation process. Mutations in genes encoding for aggregation-prone proteins, such as α-synuclein in PD and tau in AD et al., can enhance the propensity for these proteins to misfold and aggregate. For instance, studies have shown that certain genetic mutations of α-synuclein, such as H50Q and E46K, lead to changes in protein structure, resulting in the formation of new aggregate morphologies and accelerating the aggregation process [82,83]. The aggregates of these mutants not only differ structurally from the wild type but also exhibit higher toxicity and stronger seeding effects in cells [82]. Moreover, research has found that certain mutations in α-synuclein, TDP-43, and tau affect their degradation efficiency in lysosomes, thereby prolonging their half-lives, increasing the steady-state concentrations of intracellular proteins, and ultimately leading to the formation of aggregates [84]. In Alzheimer’s disease research, Aβ oligomers, particularly Aβ42, can lead to early-onset familial forms of AD, as seen with specific point mutations in the Aβ peptide that modulate nucleation processes critical for aggregation [85]. In summary, genetic mutants of proteins such as α-synuclein, tau, and Aβ promote the formation of their aggregates through various mechanisms.

### 3.4. Molecular Chaperones

Molecular chaperones, including heat shock proteins (HSPs), are integral to the cellular defense mechanisms that mitigate the effects of protein misfolding and aggregation. In Parkinson’s disease, for instance, the accumulation of α-synuclein into Lewy bodies is a hallmark of the disease. Molecular chaperones such as Hsp70 and Hsp90 have been shown to prevent the misfolding and aggregation of α-synuclein, thereby offering a potential therapeutic target for modulating disease progression [86,87]. The role of chaperones extends beyond merely preventing aggregation; they are also involved in the degradation of misfolded proteins through pathways such as the ubiquitin–proteasome system and autophagy [88,89]. In conclusion, molecular chaperones are central to the maintenance of proteostasis and the prevention of protein aggregation in neurodegenerative diseases. Their ability to modulate protein folding, prevent aggregation, and facilitate the degradation of misfolded proteins makes them attractive targets for therapeutic development.

### 3.5. Metal Ion Homeostasis

In the nervous system, the imbalance of metal ion homeostasis and the formation of protein aggregates are important pathological mechanisms of neurodegenerative diseases. As previously reviewed, metal ions such as iron, manganese, copper, and zinc participate in various physiological processes in the brain, but their excess or deficiency can lead to oxidative stress, apoptosis, and neuroinflammation, resulting in neuronal death [90]. These processes are reflected in the pathological mechanisms of neurodegenerative diseases such as Parkinson’s disease, Alzheimer’s disease, amyotrophic lateral sclerosis, and Huntington’s disease [90].

### 3.6. Other Contributing Factors (Aging, Neuroinflammation, Viral Infections)

Additional factors, as previously studied, including aging [91], neuroinflammation [92], mitochondrial dysfunction [93], and viral infections [94], among others, may also impact protein aggregation.

## 4. Ubiquitin–Proteasome System (UPS) in Protein Aggregate Clearance

The ubiquitin–proteasome system (UPS) plays a pivotal role in the degradation of protein aggregates, a process that is crucial for maintaining cellular homeostasis (Figure 3). It operates through a highly specific enzymatic cascade (ubiquitin activation by E1, conjugation by E2, and then ligation by E3) that tags target proteins with a polyubiquitin chain, marking them for destruction by the 26S proteasome. Central to this system are the E3 ubiquitin ligases, which confer specificity to the ubiquitination process by recognizing and binding to specific substrate proteins. The SCF (SKP1-CUL1-F-box protein) E3 ubiquitin ligases represent the largest family of E3 ligases and are pivotal in controlling numerous cellular processes, including cell cycle progression, apoptosis, and signal transduction [95].

Dysfunction of the ubiquitin–proteasome system (UPS) is a common feature across multiple neurodegenerative diseases, although the specific mechanisms and key players involved differ significantly [96,97,98,99,100]. For instance, in Amyotrophic Lateral Sclerosis (ALS), the effectiveness of ubiquitin–proteasome system (UPS)-mediated degradation of mutant superoxide dismutase 1 (SOD1) is a subject of ongoing debate. Various E3 ligases, such as dorfin [101], NEDL1 [102], Gp78 [103], Smurf1 [104], and MITOL [105], along with co-factors like Hsp70/CHIP [106], are involved in regulating the ubiquitination and degradation of mutant SOD1. Although the toxicity of mutant SOD1 aggregates is not fully understood, inhibition of the proteasome has been shown to increase cell death in cells expressing mutant SOD1 [107]. Specific mutations in SOD1, such as G93A, have been found to impair the expression of proteasome subunits and overall UPS function [108]. In Huntington’s Disease (HD), mutant huntingtin (mHTT) serves as a substrate for the UPS, with E3 ligase HRD1 being one of the targeting agents [109]. The UPS appears to be inadequate in managing the heightened proteolytic demand caused by mHTT accumulation, suggesting that enhancing mHTT ubiquitination and subsequent proteasomal degradation could be a viable therapeutic approach. In Parkinson’s Disease (PD), several key proteins associated with the disease, including α-synuclein, parkin, PINK1, and UCH-L1, have strong links to the UPS. Parkin, an E3 ligase, facilitates mitophagy by ubiquitinating mitochondrial proteins such as Mfn1 and Mfn2 [110]. The PINK1–Parkin pathway plays a crucial role in regulating the degradation of Znf746, thereby offering protection to dopaminergic neurons [111]. Proteasome dysfunction (e.g., depletion of subunits like Rpt2) directly causes α-synuclein aggregation, Lewy body-like inclusion formation, and neurodegeneration in models [112]. In Alzheimer’s Disease, UPS dysfunction is a hallmark of AD, contributing to both Aβ and tau pathologies through impaired degradation and altered function of key regulators (Parkin [113], UCHL1 [114], CHIP [115]). Restoring UPS function/E3 ligase activity represents a potential therapeutic avenue.

## 5. Autophagy in Protein Aggregates Clearance

### 5.1. Selective Autophagy and Aggrephagy

Autophagy, a cellular degradation process, as previously reviewed, plays a crucial role in maintaining cellular homeostasis by removing damaged organelles and proteins. It is implicated in various physiological and pathological processes, including neurodegeneration, cancer, cardiovascular diseases, and musculoskeletal disorders [116]. This process is categorized into three main types: macroautophagy, microautophagy, and chaperone-mediated autophagy, each differing in their mechanisms of cargo delivery to lysosomes. Macroautophagy, the most extensively studied form, involves the formation of double-membrane vesicles known as autophagosomes that sequester cytoplasmic material and deliver it to lysosomes for degradation [117]. This process is highly regulated and involves the recognition and sequestration of specific cargo by autophagy receptors, which play a pivotal role in cargo specificity and autophagosome formation. The interaction between autophagy receptors and ubiquitin-like proteins, such as Atg8/LC3, forms the molecular basis for selective autophagy, ensuring the recognition and removal of various cytosolic cargoes, including aggregated proteins, damaged organelles, and pathogens [118]. The selective nature of autophagy is exemplified by the role of cargo receptors like p62/SQSTM1, which facilitate the degradation of ubiquitinated cargo and are crucial for maintaining cellular proteostasis [119]. The process of autophagosome maturation and fusion with lysosomes is a key step in autophagy, mediated by various proteins and complexes. Myosin VI, in conjunction with its adaptor proteins NDP52, optineurin, T6BP, and Tom1, plays a crucial role in autophagy by facilitating the delivery of endosomal membranes to autophagosomes, promoting their maturation and subsequent fusion with lysosomes [120].

The ubiquitin–proteasome system (UPS) and autophagy are the two major protein degradation pathways in cells. The UPS is primarily responsible for the degradation of short-lived and misfolded proteins, but its capacity can be exceeded in neurodegenerative conditions, leading to the accumulation of toxic protein aggregates. When UPS is impaired, autophagy, a lysosome-based degradative process, becomes indispensable for the clearance of these aggregates, thereby preventing cellular toxicity and neurodegeneration [121,122] (Figure 3). Autophagy is particularly vital in the context of neurodegenerative diseases, where the accumulation of protein aggregates is a hallmark feature [123,124]. The selective degradation of protein aggregates through autophagy, often referred to as aggrephagy, involves the recognition and clearance of these aggregates by specific receptors and adaptors, such as p62, NBR1, and CCT2, which facilitate their delivery to the lysosome for degradation [125,126,127,128].

### 5.2. Key Autophagy Receptors and Mechanisms

The primary aggrephagy cascade in eukaryotes is initiated through the coordinated actions of the cargo receptors p62 (also referred to as SQSTM1), the sequestosome-like receptor (SLR) NBR1 and TAX1BP1 [129,130]. The principal selective trigger for this pathway is the ubiquitylation of cargo proteins. Notably, ubiquitin-independent pathways that rely on p62 have also been identified, some of which are activated by N-terminally arginylated proteins [131]. Additionally, another type of ubiquitin-independent pathway has been documented, which focuses on the removal of aggregates via a different SLR, optineurin [132]. Both p62 and NBR1 possess UBA domains that recognize ubiquitin, as well as PB1 domains that facilitate homo- and hetero-oligomerization [133,134]. In p62, the PB1 domain enables oligomerization and interaction with the PB1 domain of NBR1. In contrast, NBR1 is incapable of oligomerizing via its PB1 domain due to the absence of one of the oligomerization interfaces [130]. The p62-NBR1 hetero-oligomer, likely to consist of multiple p62 molecules capped by NBR1, is capable of sequestering ubiquitylated proteins within larger condensates [134]. Following the formation of p62-NBR1-dependent condensates, several additional factors contribute to the facilitation of autophagy. One of the initial factors recruited to these condensates is TAX1BP1, which interacts with NBR1 [130]. Subsequently, TAX1BP1 promotes the initiation of autophagosome biogenesis at the cargo site by recruiting the core autophagy protein and ULK1 complex subunit FIP200 (also referred to as RB1CC1) through its SKICH domain [130]. As the nascent autophagosomal membrane expands around cargo, it becomes adorned with LC3 proteins. These proteins interact with the LC3-interacting region (LIR) motifs found in the cargo receptors p62, NBR1, and TAX1BP1. This final step facilitates the tethering of the cargo to the expanding membrane. Upon the completion of autophagosome biogenesis, the cargo is encapsulated within a closed double membrane and subsequently directed to the lysosome. Therefore, cargo receptors play a pivotal role in aggrephagy, as they are integral to condensing the cargo, recruiting the machinery necessary for the initiation of autophagosome biogenesis, and tethering the cargo to the membrane. Furthermore, recent studies have identified the chaperonin subunit CCT2 as an aggrephagy receptor responsible for regulating the clearance of solid protein aggregates, operating independently of ubiquitin-binding receptors [126,127].

### 5.3. Neuronal Autophagy and Pathological Alterations

Under physiological conditions, the majority of a neuron’s catabolic lysosomal activity occurs within the soma [135]. However, autophagy is also active at synapses, necessitating the evolution of systems to transport mature autophagosomes from distal projections to these sites [136]. The condensation effect of cargo receptors is likely crucial for the initial sequestration of misfolded proteins located far from the neuronal degradative zones under basal conditions. In this context, the protein p62 has been identified as playing a protective role in counteracting protein aggregation [137]. Similarly, ALFY has been reported to have a protective function in delaying the onset of Huntington’s disease [138]. The dysfunction of aggrephagy has been implicated in the pathogenesis of several neurodegenerative disorders, including Alzheimer’s disease (AD), Parkinson’s disease (PD), amyotrophic lateral sclerosis (ALS), and HD (Huntington’s disease) where the accumulation of misfolded proteins leads to cellular toxicity and neuronal death [123,124,137,138,139,140]. While our understanding of how aggrephagy safeguards the neuronal proteome is advancing, further research is required to elucidate the temporal and spatial coordination of cargo condensation, autophagosome biogenesis, and lysosomal degradation.

Under pathological conditions, neurons experience significant alterations in their autophagic flux, characterized by a general reduction in degradative capacity. The accumulation of autophagic vesicles is notably a hallmark of Alzheimer’s Disease (AD) [141]. Additionally, autophagy is aberrantly upregulated in axonal spheroids, which are pathological structures identified in AD-affected brains and are classified as dystrophic neurites [142]. The disruption of aggrephagy is apparent from the onset of the aggrephagy cascade; for instance, p62 accumulates and colocalizes with Tau and α-synuclein amyloid aggregates in AD and Parkinson’s Disease (PD), respectively [140,143]. Although additional aggregate-specific autophagy pathways, mediated by UXTl (also known as TRADD) and CCT2 [125,127], have been identified, protein aggregation ultimately predominates in various diseases. One possible explanation is that, at a certain point, the aggrephagy cascade may fail to recruit the downstream components necessary for autophagosome biogenesis. Alternatively, and not mutually exclusively, the burden of aggregation may surpass the degradative capacity of the lysosomal system. Collectively, components of the aggrephagy process play an essential role in maintaining the balance of the neuronal proteome and are disrupted during neurodegeneration. Restoring autophagy has emerged as a promising therapeutic strategy to combat neurodegenerative diseases, which are often characterized by the accumulation of misfolded proteins and dysfunctional organelles. For instance, the PINK1/Parkin pathway is critical for mitophagy, and its activation has been shown to protect against PD-related neurodegeneration [144]. Enhancing autophagic activity can potentially mitigate the progression of these diseases by promoting the clearance of toxic protein aggregates and restoring neuronal function [145].

## 6. Role of Molecular Chaperones in Protein Aggregates Clearance

### 6.1. Chaperone Functions in Protein Folding and Clearance

Molecular chaperones are pivotal in maintaining cellular proteostasis, particularly in the context of neurodegenerative diseases where protein misfolding and aggregation are prevalent. These chaperones, as previously reviewed, including heat shock proteins (HSPs), play a crucial role in assisting the proper folding of nascent proteins, refolding misfolded proteins, and targeting irreversibly damaged proteins for degradation [146,147]. The importance of molecular chaperones is underscored by their role in preventing protein aggregation, a process implicated in numerous neurodegenerative diseases such as Alzheimer’s and Parkinson’s diseases [146,147].

The structural basis of chaperone function is exemplified by the trigger factor (TF) chaperone, which prevents protein aggregation through multivalent binding to unfolded polypeptides, thereby maintaining them in an extended conformation and preventing premature folding [148]. Chaperones like Hsp70 and Hsp90 are particularly noteworthy for their collaborative roles in protein remodeling and degradation pathways. These chaperones, along with their co-chaperones, facilitate the folding and refolding of proteins, and are involved in the ubiquitin–proteasome system and autophagy, which are essential for the degradation of misfolded proteins [149,150,151]. Moreover, the ability of chaperones to interact with and neutralize toxic oligomers, as seen with DNAJB6’s inhibition of amyloid-β aggregation, exemplifies their critical role in mitigating the cytotoxic effects of protein aggregates [152,153].

### 6.2. Chaperone-Mediated Autophagy (CMA)

Chaperone-mediated autophagy (CMA) is a highly selective form of autophagy that plays a crucial role in maintaining cellular homeostasis by degrading specific cytosolic proteins. This process is distinct from other autophagic pathways due to its selectivity and the direct translocation of substrate proteins across the lysosomal membrane. CMA is facilitated by the lysosome-associated membrane protein type 2A (LAMP-2A) and the heat shock cognate protein 70 (HSC70), which recognize and transport proteins containing a KFERQ-like motif to the lysosome for degradation [89,154] (Figure 3). As previously examined, the selectivity of CMA allows for the precise regulation of protein turnover, which is essential for various cellular processes, including metabolism, immune responses, and stress adaptation [155,156]. The loss of CMA function has been shown to exacerbate proteotoxicity and accelerate neurodegenerative disease progression, as evidenced in Alzheimer’s disease models [89]. Enhancing CMA activity has been demonstrated to ameliorate disease pathology, highlighting its potential as a therapeutic target [89,157]. Moreover, a study utilizing a nanochaperone-mediated autophagy approach showed that targeting pathogenic tau with customized nanochaperones can enhance autophagic flux and clear tau aggregates, significantly alleviating tau burden and cognitive deficits in AD mouse models [158]. Similarly, in Parkinson’s disease, the degradation of alpha-synuclein via CMA is critical for preventing its aggregation, which is implicated in the formation of Lewy bodies and subsequent neurodegeneration. Studies have shown that wild-type alpha-synuclein is degraded by both CMA and macroautophagy, with CMA playing a significant role in neuronal cells [159]. The dysfunction of CMA, often due to genetic mutations or environmental factors, leads to the accumulation of alpha-synuclein, contributing to PD pathogenesis [160,161,162]. In summary, molecular chaperones are integral to the maintenance of proteostasis and the prevention of neurodegenerative diseases. Their ability to modulate protein folding, aggregation, and degradation pathways positions them as promising therapeutic targets.

### 6.3. Dual Roles of Chaperones in Aggregation and Protection

However, recent studies have highlighted scenarios where chaperones may inadvertently contribute to protein aggregation, particularly under stress conditions or when overwhelmed by misfolded proteins. For instance, under conditions of proteostasis stress, such as those induced by genotoxic agents, the chaperone system can become overloaded, leading to increased aggregation of a metastable subproteome [163]. This suggests that while chaperones are essential for preventing aggregation, their capacity can be exceeded, resulting in unintended aggregation. Interestingly, some chaperones have been shown to facilitate the aggregation of misfolded proteins as a protective mechanism. For example, the small heat shock protein Hsp42 in yeast promotes the sequestration of misfolded proteins into cytosolic aggregates, which may serve as a cytoprotective strategy by isolating potentially toxic species [164]. Similarly, during heat stress, chaperones can facilitate the formation of protein aggregate centers (PACs), which protect thermo-sensitive proteins from degradation and allow for their eventual disassembly and refolding [165]. These findings indicate that chaperone-mediated aggregation can be a regulated process aimed at minimizing cellular damage.

## 7. Diagnostic Techniques for Protein Aggregates

### 7.1. Imaging Modalities

Imaging modalities play a crucial role in the detection of protein aggregates in neurodegenerative diseases. Magnetic resonance imaging (MRI) can provide information about brain structure and function. For example, in Alzheimer’s disease, MRI can detect atrophy in specific brain regions, such as the hippocampus, which is associated with the accumulation of protein aggregates [166]. Positron emission tomography (PET) is another important imaging technique. PET tracers can be used to detect the presence of specific protein aggregates, such as Aβ plaques in AD. Compounds like [^11^C]Pittsburgh compound B can bind to Aβ fibrils, allowing for in vivo visualization of plaque deposition [167]. In Parkinson’s disease, PET imaging can also be used to assess the integrity of the dopaminergic system and detect abnormal protein aggregation-related changes [167]. Moreover, retinal hyperspectral imaging (rHSI) is an emerging non-invasive technique. It can detect early pathological changes in the retina related to AD, potentially due to the presence of low-order Aβ aggregates. The technique shows promise for the early detection of AD, as it can be conducted non-invasively without the need for exogenous labels [168].

### 7.2. Biomarkers in Cerebrospinal Fluid and Blood

Biomarkers are also valuable for the early diagnosis, prognosis, and monitoring of neurodegenerative diseases associated with protein aggregates. In Alzheimer’s disease, Cerebro-Spinal Fluid (CSF) biomarkers such as Aβ42, phosphorylated tau (p-tau), and total tau (t-tau) are well-studied. As previously examined, decreased levels of Aβ42 and increased levels of p-tau and t-tau in the CSF are associated with the presence of Aβ plaques and neurofibrillary tangles in the brain [169]. In Parkinson’s disease, α-synuclein in the CSF or blood may serve as a biomarker. Although its measurement is still challenging, some studies have shown that levels of α-synuclein or its modified forms may be related to disease progression [170,171,172]. Additionally, other proteins such as UCHL1 and GPNMB in the CSF have been identified as candidate biomarkers for ALS, which shares some molecular deficits with PD, including disruption of protein homeostasis [173]. Extracellular vesicles have also emerged as a potential source of biomarkers. They can carry pathological protein aggregates such as Aβ, tau, and α-synuclein, and their cargo may reflect the pathological changes in the brain [54,174,175,176].

### 7.3. Molecular and Genetic Diagnostics

Advances in molecular diagnostics have provided new tools for detecting protein aggregates. Recent advancements in reporter assays, particularly those employing Nano-luciferase, offer a novel approach for detecting translational errors. These errors may be associated with the formation of protein aggregates observed in neurodegenerative diseases [177]. Proteomics techniques, including targeted multiple reaction monitoring (MRM) mass spectrometry, can rigorously quantify proteins from known disease genes involved in lysosomal, ubiquitin-proteasomal, and autophagy pathways in the CSF [178,179]. Genetic analysis also plays a role. For example, the identification of specific gene mutations, such as the GGGGCC repeat expansion in c9orf72 gene associated with frontotemporal lobar dementia and ALS, can help in the diagnosis and understanding of the disease mechanism related to protein aggregation [180]. This has helped identify candidate biomarkers for diseases like ALS and PD, such as changes in the levels of chromogranin B, UCHL1, GPNMB, and cathepsin D. In summary, the identification and differentiation of co-pathologies through specific biomarker profiles is an emerging area of interest in the field of biomedical research. The complexity of diseases such as Alzheimer’s disease (AD) and other age-related conditions necessitates the development of precise diagnostic tools that can distinguish between overlapping pathologies.

## 8. Therapeutic Strategies for Protein Aggregate Degradation

### 8.1. Pharmacological Approaches

Pharmacological approaches aim to enhance the clearance of protein aggregates through various mechanisms. Some compounds can directly interact with misfolded proteins to prevent their aggregation. For example, tetrahydrofolic acid (THF) has been shown to accelerate the fibrillation of α-synuclein and amyloid-β protein (Aβ), reducing the abundance of cytotoxic oligomers and suppressing their toxicity [181]. Other drugs target the cellular pathways involved in protein aggregate clearance. Cannabidiol, for instance, can induce autophagy via the CB1 receptor and reduce cytosolic α-synuclein levels in a neuroblastoma cell line [182]. This suggests that cannabinoid compounds acting on the CB1 receptor could be a new approach for modulating autophagy and degrading protein aggregates. Hydroxyl chalcone derivative DK02 has shown promise as a multi-target-directed ligand for Alzheimer’s disease. In a zebrafish model, it enhanced cognitive functions, reduced oxidative stress—induced BACE1 expression, and decreased tau phosphorylation at disease—relevant sites [183]. Phthalocyanines, for instance, have been shown to inhibit the aggregation process and neutralize the associated toxicity of amyloidogenic proteins like alpha-synuclein, tau, and amyloid-β. These compounds modulate amyloid assembly through specific interactions with aromatic residues in the target proteins, offering a promising tool for developing therapeutic strategies against neurodegenerative disorders [184]. A recent study showed that aspirin has been found to promote the clearance of alpha-synuclein aggregates through K63-linked ubiquitination, providing insights into its diverse pharmacological effects and potential as a therapeutic agent [185]. Additionally, the therapeutic potential of modulating autophagy in these diseases has garnered significant interest, with research focusing on both the activation and inhibition of autophagy pathways to restore cellular homeostasis and mitigate disease progression [186]. Small molecules such as USP30 inhibitors and PINK1 activators are being explored in clinical trials to enhance mitophagy, offering a novel therapeutic strategy [187,188]. Additionally, as previously studied, the role of the AMPK pathway in promoting autophagic flux by modulating mitochondrial dynamics has been highlighted as a potential therapeutic target, particularly in prion diseases [189].

### 8.2. Gene Therapy

Gene therapy techniques offer potential strategies to target protein aggregates in neurodegenerative diseases. In Huntington’s disease, viral vector-mediated approaches have been used to model the key neuropathological features of the disease, such as the production of abnormal intracellular protein aggregates [190]. These models can help in understanding the disease mechanism and testing new therapeutic interventions. For dominant muscle diseases like myotonic dystrophy, antisense oligonucleotides have been developed to target the mutant RNAs containing expanded repeats. These RNAs form aggregates and alter the activities of alternative splicing regulators. Antisense oligonucleotides can promote the degradation of the expanded CUG transcripts, which is a promising therapeutic strategy [191]. Furthermore, recent article indicates targeted protein degradation (TPD) technologies, such as the ubiquitin–proteasome and autophagy–lysosome pathways, have been explored as means to clear misfolded proteins from cells. These pathways can be harnessed through gene therapy to enhance the degradation of toxic protein aggregates, thereby alleviating their pathological effects [192]. Additionally, gene therapy can be used to upregulate the expression of molecular chaperones, such as heat shock proteins, which assist in the proper folding of proteins and prevent aggregation [193].

### 8.3. Immunotherapy

Immunotherapy has emerged as a potential treatment strategy for neurodegenerative diseases associated with protein aggregates [194,195]. Antibodies can be designed to target specific protein aggregates, such as Aβ in Alzheimer’s disease or α-synuclein in Parkinson’s disease. These antibodies can potentially promote the clearance of aggregates by mechanisms such as enhancing phagocytosis by immune cells. In some cases, the immune system can recognize and respond to misfolded proteins. However, in neurodegenerative diseases, the normal immune response may be insufficient or dysregulated. Immunotherapy aims to boost this response. For example, as previously reviewed, active immunization with Aβ peptides has been tested in clinical trials, with the hope of inducing the production of antibodies that can clear Aβ plaques [196]. Another approach is the use of intracellular antibodies. Antibodies that gain access to the cytoplasm can bind to protein aggregates, and in some cases, the antibody–protein complex can be targeted for degradation by the proteasome or autophagy pathways [197]. The blood–brain barrier (BBB) is a critical physiological structure that serves as a gatekeeper, regulating the transport of molecules between the bloodstream and the central nervous system (CNS). This barrier, while essential for protecting the brain from toxins and pathogens, poses a significant challenge for the delivery of therapeutic agents to treat neurological and neurodegenerative diseases. To address this, novel strategies such as bispecific antibodies and engineered exosomes are being developed to facilitate the delivery of immunotherapeutics across the BBB [198,199]. A significant milestone has been the recent FDA approval of the first disease-modifying antibodies for Alzheimer’s disease, which validate the amyloid hypothesis and the therapeutic potential of immunotherapy. Lecanemab (Leqembi^®^) and Donanemab (Kisunla^®^) are prime examples; these monoclonal antibodies selectively target and facilitate the removal of protofibrils and insoluble, deposited amyloid-β plaques, respectively.

### 8.4. Nanoparticles

The restoration of lysosomal acidity and activity through nanoparticle interventions holds significant promise for addressing neurodegenerative diseases such as Parkinson’s Disease (PD) and Alzheimer’s Disease (AD). One promising approach involves the use of biodegradable nanoparticles, such as poly (lactic-co-glycolic acid) (PLGA) nanoparticles, which have been shown to restore lysosomal pH and autophagic flux in lipotoxic pancreatic beta cells. These nanoparticles deliver lactic and glycolic acid to lysosomes, effectively lowering lysosomal pH and improving cellular degradative activity [200]. Researchers found that restoring lysosomal acidity using targeted nanoparticles improved mitochondrial turnover and bioenergetic function. This finding is particularly relevant to neurodegenerative diseases, where mitochondrial dysfunction is a common feature. By enhancing lysosomal function, nanoparticles could indirectly support mitochondrial health, thereby preserving neuronal function and viability in PD and AD [201].

## 9. Controversies and Future Directions in Protein Aggregates Research

### 9.1. Debates on the Toxicity of Protein Aggregates

Debates on the Toxicity of Protein Aggregates: One of the main debates in protein aggregate research is regarding the toxic species. While it was initially thought that mature amyloid fibrils were the main toxic agents in neurodegenerative diseases, increasing evidence suggests that soluble oligomeric forms of proteins may be more toxic [26,202,203]. The mechanism of toxicity is also a matter of debate. Some studies suggest that protein aggregates may disrupt normal cellular functions by interfering with key cellular processes such as membrane integrity, ion channels, and intracellular signaling pathways [204]. Others propose that the aggregates may sequester important cellular factors, leading to cellular dysfunction and death [205]. There is also controversy regarding the role of protein aggregates in the disease process. Some view them as the primary cause of neurodegeneration, while others consider them to be a secondary consequence of other underlying pathological processes [206].

### 9.2. Emerging Technologies for Aggregate Clearance

Emerging Technologies for Protein Aggregate Clearance: Emerging technologies offer new hope for the clearance of protein aggregates. New autophagy receptors are being identified, such as CCT2 [127], which specifically promotes the autophagic degradation of solid protein aggregates. CCT2 associates with aggregation-prone proteins independent of cargo ubiquitination and interacts with autophagosome markers through a non-classical VLIR motif [127]. Nanoparticles are also being explored for their potential in protein aggregate clearance. Europium hydroxide [Eu (III)(OH)3] nanorods, for example, can reduce huntingtin protein aggregation by inducing autophagy in different cell lines [207]. These nanorods can enhance the expression of autophagy marker protein LC3-II and degrade the selective autophagy substrate p62/SQSTM1. MnFe_2_O_4_ nanoparticles accelerate the clearance of mutant huntingtin selectively through the ubiquitin–proteasome system [208]. Targeted protein degradation (TPD) technologies, including proteolysis targeting chimeras (PROTACs), molecular glues, and lysosome-targeted chimeras (LYTACs), emerging as promising strategies. These small-molecule-based approaches can selectively degrade target proteins, potentially including those involved in protein aggregate formation [209].

### 9.3. Future Research Priorities

As previously reviewed, future research in protein aggregates is likely to focus on a better understanding of the complex mechanisms underlying protein misfolding, aggregation, and clearance. This may involve the use of advanced imaging techniques, such as in-vivo imaging with higher resolution, to visualize the formation and spread of protein aggregates in real time [210]. The development of more specific and effective biomarkers will also be crucial. These biomarkers can help in the early diagnosis of neurodegenerative diseases, as well as in monitoring disease progression and treatment response [211].

## Figures and Tables

**Figure 1 ijms-26-10568-f001:**
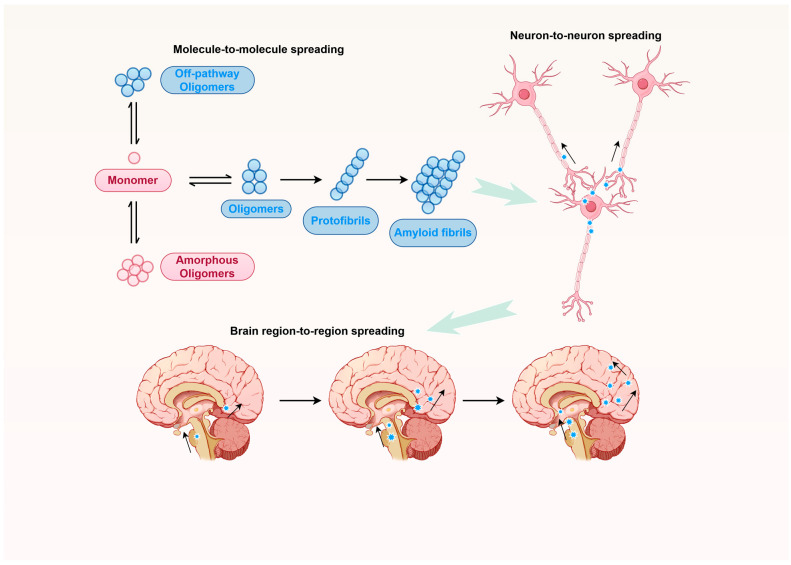
Protein aggregation and the principle of pathological transmission. The transmission of biological information via prions by seeding of protein aggregation operates at multiple levels: At the molecular scale, the template-induced transformation of a natively folded protein (red color) by a polymeric misfolded protein result in the autocatalytic expansion of protein aggregates (blue color). The arrows represent the template-directed misfolding and incorporation of native proteins into the growing aggregate. At the cellular scale, the pathology propagates from one cell to another via the transfer of misfolded protein aggregates (blue stars) between neighboring cells, resulting in the regional dissemination of abnormalities. The arrows indicate the release of aggregates from a donor cell, their transit through the extracellular space, and subsequent uptake/internalization by a recipient cell, where they act as seeds to nucleate new aggregation. At the organ scale, the progressive spread of pathology (blue stars) among cells culminates in tissue damage, which can extend to remote or distant brain regions through mechanisms such as cell-to-cell contact or the movement of biological fluids, including interstitial fluid, cerebrospinal fluid, or blood. The arrows depict the directionality and routes of pathological spread throughout the brain network, leading to the characteristic progression of neuroanatomical involvement in these diseases.

**Figure 2 ijms-26-10568-f002:**
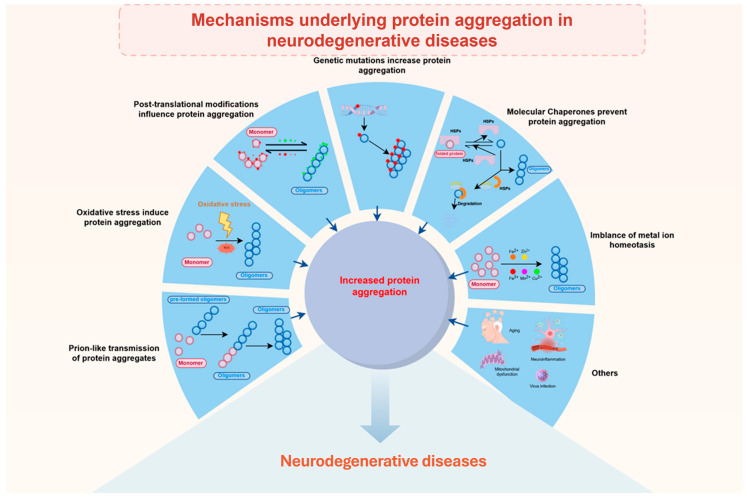
Mechanisms and modulating factors of toxic protein aggregation in neurodegenerative diseases. The schematic illustrates the multi-step process of protein aggregation and the cellular factors that influence it. A native monomeric protein (red color) can misfold and self-assemble into soluble oligomers (blue color), which further develop into insoluble amyloid fibrils and large aggregates. This pathogenic cascade is influenced by several key mechanisms (as indicated by black arrows): Genetic mutations (symbolized by a DNA double helix, representing gene variants that increase aggregation propensity) and post-translational modifications (represented by chemical group icons, such as stars, pentagrams, etc. And the green color denotes modifications that facilitate oligomers formation, whereas the red color signifies modifications that impede oligomers formation.) can directly regulate aggregation. Cellular stressors, such as oxidative stress (depicted as a lightning bolt) and imbalance of metal ion homeostasis, also accelerate this process. Molecular chaperones help maintain proteostasis by preventing protein aggregation under normal conditions; however, their impaired function or overload can contribute to aggregation. Other factors, such as aging, inflammation, mitochondrial dysfunction, virus infection et al. could also influence protein aggregation. The net imbalance between these aggravating and protective factors leads to increased accumulation of neurotoxic protein aggregates, a hallmark of various neurodegenerative diseases.

**Figure 3 ijms-26-10568-f003:**
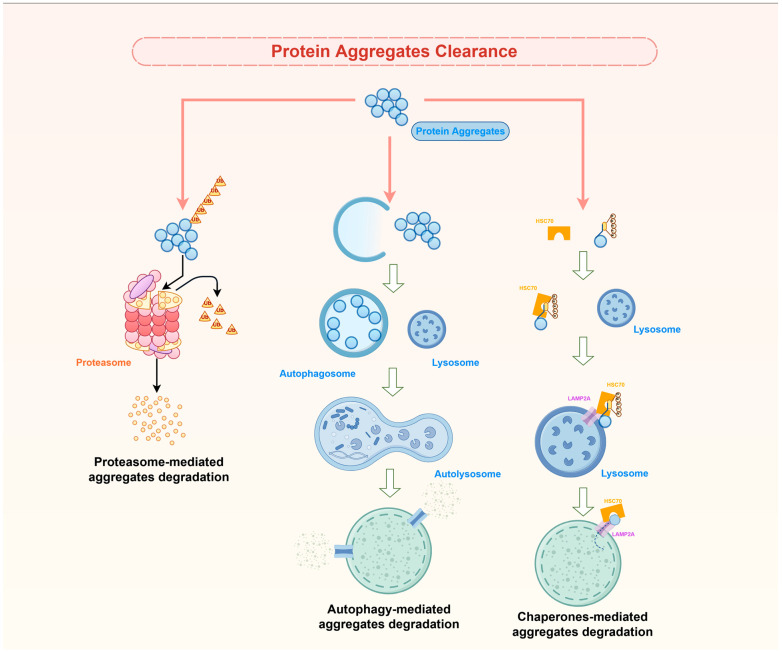
Major pathways of protein aggregation clearance. The schematic illustrates the three primary cellular systems responsible for clearing toxic protein aggregates. Protein aggregates to be targeted for degradation by the proteasome are recognized by poly-ubiquitin chain of four or more ubiquitins. Then, the ubiquitin chain is released, and the aggregate proteins are degraded into small peptides by the 26S proteasome. Finally, the ubiquitin is recycled by release of free monomeric ubiquitin from the ubiquitin chain. The arrows illustrate the process of ubiquitinated aggregates being recognized and delivered to the 26S proteasome, where the polyubiquitin chain is identified by receptors on the 19S regulatory cap; Autophagy sequesters protein aggregates by a delimiting membrane that forms through conjugation of specific proteins among themselves and with lipids in a complex multistep process. The membrane then seals into an autophagosome that is trafficked by microtubules. Fusion of autophagosomes with lysosomes mediates degradation of the trapped aggregates. Arrows indicate the each steps of the autophagy: Phagophore Formation, Sequestration of Protein Aggregate, Vesicle Trafficking, Fusion with Lysosome and Degradation in Lysosome; Chaperone-mediated autophagy (CMA) involves the selective degradation of KFERQ-like motif-bearing aggregates delivered to the lysosomes via chaperone HSC70, and their internalization in lysosomes via the receptor lysosome-associated membrane protein type 2A (LAMP2A). Arrows indicate the each steps of CMA: substrate recognition and binding, targeting substrate to the lysosome, aggregated proteins being degraded inside the Lysosome.

**Table 1 ijms-26-10568-t001:** Major Protein Aggregates and Associated Diseases.

Protein Name	Primary Associated Disease(s)	Aggregate Morphology & Key Characteristics	Common Abbreviation	Reference
Amyloid-β	Alzheimer’s disease (AD)	Forms extracellular senile plaques (SPs). Aβ is produced by the sequential cleavage of the amyloid precursor protein (APP) by β- and γ-secretases. The Aβ42 isoform is more aggregation-prone and toxic than Aβ40.	Aβ	[27]
Tau	Alzheimer’s Disease (AD), Pick’s Disease (PiD), Progressive Supranuclear Palsy (PSP), Corticobasal Degeneration (CBD), Argyrophilic Grain Disease (AGD), Chronic Traumatic Encephalopathy (CTE)	Forms intracellular neurofibrillary tangles (NFTs) in AD, Pick Bodies in PiD, NFTs, Tufted Astrocytes, Coiled Bodies in PSP, Astrocytic Plaques, Neuronal tau inclusions, Coiled Bodies in CBD, Argyrophilic Grains, Coiled Bodies in AGD, Perivascular neuronal tau deposits, and clustering in sulcal depths in CTE. Tau normally stabilizes microtubules. In disease, it becomes hyperphosphorylated, misfolds, and aggregates into fibrils.	tau	[28]
α-synuclein	Parkinson’s Disease (PD), Dementia with Lewy Bodies (DLB), Multiple System Atrophy (MSA)	Forms neuronal Lewy Bodies in PD/DLB, glial cytoplasmic inclusions (GCIs) and oligodendrocytes in MSA. Its aggregation involves a process from monomers to oligomers to protofibrils and finally mature fibrils.	α-syn, aSyn	[29]
Huntingtin	Huntington’s Disease (HD)	Contains an expanded polyglutamine (polyQ) tract, leading to protein misfolding, aggregation, and the formation of intracellular inclusions.	HTT, Htt	[30]
TAR DNA-binding protein 43	Amyotrophic Lateral Sclerosis (ALS), Frontotemporal Lobar Degeneration (FTLD)	Normally located in the nucleus, it mis-localizes to the cytoplasm in disease states, where it aggregates and forms inclusions.	TDP-43	[31]

## Data Availability

No new data were created or analyzed in this study. Data sharing is not applicable to this article.

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
