# Peer review of "Current Understanding of Protein Aggregation in Neurodegenerative Diseases"

_ijms, 2025, doi:10.3390/ijms262110568_

Round 1

Reviewer 1 Report

Comments and Suggestions for Authors

See attached file.

In the manuscript “Current Understanding of Protein Aggregation in Neurodegenerative Diseases” by Chen Hu et al., an interesting review of protein aggregation focused on neurone disease is conducted by a team of Shandong University of Technology. Here the idea is to cover the topic from the molecular basis to the diagnosis and future treatment. This choice is particularly interesting because it offers the reader a general overview of the problem of the pathogenicity of aggregated proteins. However, the document has major flaws that could be corrected.

Major comment

A very high number of citations refers to reviews. This process hinders the reader in finding the articles that are the source of the idea developed or the result presented. When their reference is a review, authors should absolutely specify in the text that they cite a review: “as previously reviewed” or something like that. Moreover, the systematic use of review is not acceptable, they should therefore look for the references of experimental articles rather. Nobody wants to read a review of reviews.

Chap2 “Historical perspectives on protein aggregation”: with a title like that, the authors should specify the first article describing this pathology, the first article describing the molecular bases of the disease. Also line 62, beta-sheet importance is too briefly presented (by a review…). Please use original article for reference and develop the question of beta sheet.

On page 2 and 3, authors introduce the concept of “prion like property”, they should explain how this new and key concept was demonstrated.

On page 5, the PTM paragraph does not cite enough references to research articles. In addition, the isomerization of aspartic acid is neither presented nor discussed. An additional paragraph regarding protein aggregation and isomerization should be added next to or within the PTM paragraph.

Line 168, the word "degron" is used in be explained? what are the characteristics of these regions? motifs? enrichment in certain amino acids? etc.

In chapter 5 “autophagy”, authors should explain shortly cargo, cargo protein and cargo receptor and the relation with lysosome at the beginning of the chapter.

In Chapter 6, the authors present chaperone proteins with only a positive cleaning function. Are they sure that these proteins do not have a negative function? Can we assume that the uptake of aggregated proteins and their internalization into vesicles does not contribute to the propagation of the pathology? Along the same lines, a paragraph on cell-to-cell propagation would be of major interest

In chapter 8 “therapeutic”, blood-brain barrier should be explained in the context of immunotherapy.

Reviewer 2 Report

Comments and Suggestions for Authors

Hu et al. gave a broad overview of the current knowledge and controversies on the role of protein aggregation in different neurodegenerative diseases, starting with the different mechanisms involved in their formation, followed by the cellular pathways responsible for their clearance and their dysfunction in neurodegenerative diseases, and concluding by discussing the potential of targeting these aggregates for clinical purposes as biomarkers, for diagnosis as well as therapeutic strategies.

This review is overall well written but I think the authors tried to discuss too many issues in one review, which makes each section a bit descriptive. Whether the authors have any expertise or track record in the field of protein aggregation and neurodegeneration is not apparent at all in the review. 

Specific issues:

  • Some redundancies here and there in the text should be revised to further improve the text: sections 2 and 3; paragraph from l.96 to 103; paragraph on molecular chaperones (l.198 to 215) is redundant with part 6.
  • Some paragraphs and statements miss references (section 4 particularly)
  • Section 2 should be renamed and/or merged with section 1 (intro) as this paragraph is not really a historical perspective
  • Paragraph l.62 to 64: maybe some examples for each type
  • Figure 1: not super accurate
    • neuron-to-neuron scheme: not clear which form is involved in the spreading
    • brain region-to-brain region scheme: the arrows are misleading, we can't understand if the pathogenesis is starting at the periphery of the brain or from the brain stem
  • Paragraph l.72 to 80: information regarding the localization of these aggregates (extracellular vs intracellular) should be provided. Also prions do not 'induce the misfolding of proteins' but rather shift the equilibrium between folding intermediates towards aggregation.
  • Paragraph on the propagation mechanisms (l.103) should mention other pathways than free aggregates, such as transmission via extracellular vesicles.
  • Paragraph on the heterogeneity of these aggregates (l.106): include a word about the co-occurence of different aggregates in some diseases
  • Section 3: use subsections to make this section clearer
  • 190-194: the existence of  Abeta*56 is highly controversial. This should be mentioned with proper references.
  • Section 4: Maybe a brief general description of the UPS mechanisms could be interesting, specifically to introduce the E3 ligases
  • Figure 3: ubiquitin is also important for autophagy pathway
  • Section 5: introduce the term macroautophagy
  • 268: unwanted proteins (toxic proteins, aggregates)
  • At the end of the section 5: maybe introduce the concept of pathological loops with examples and a word about restoring autophagy to prevent neurodegenerative diseases
  • Section 7: exosomes should be extracellular vesicles
  • 420 to 421: not clear
  • At the end of the section 7: a word on the need to have specific biomarkers profile to differentiate co-pathologies
  • The paragraph on pharmacological approaches is not exhaustive enough; what about all the studies on activators/inhibitors of autophagy?
  • Paragraph on immunotherapy: maybe give as examples the last antibodies accepted by FDA for AD?
  • Paragraph on nanoparticles: maybe cite also studies on nanoparticles that restore lysosome acidity/activity in PD or AD

Reviewer 3 Report

Comments and Suggestions for Authors

Several neurodegenerative disorders, including Alzheimer's disease, Parkinson's disease, and Amyotrophic Lateral Sclerosis, although displaying quite different clinical symptoms, all share a commonality: aggregates of misfolded proteins, both intracellularly and extracellularly, at the level of the central nervous system. Protein aggregates exhibit a form of toxicity towards cells, impairing neuronal functions and ultimately leading to cell death. Remarkably, protein aggregates exhibit prion-like properties, thus facilitating their propagation across neural networks and exacerbating the progression of neurodegenerative diseases.

The authors of the manuscript titled "Current Understanding of Protein Aggregation in Neurodegenerative Diseases" examined the roles of protein aggregates in neurodegenerative disorders.

Overall, the manuscript offers interesting hints, and no major issues are detected. However, before publication, a couple of minor amendments are required.

  1. Paragraph 8 "Therapeutic strategies for protein aggregates degradation" should be smoothed a little bit. Apparently, preclinical data are missing or quite poor. Additionally, whenever available, they should be delivered to the central nervous system, facing a serious hurdle, namely, the blood-brain barrier.
  2. A few spaces are missing, especially when reference citations appear within the main text, and a few typos seem scattered throughout the main text (e.g., line 230).
  3. For consistency, names should be harmonized throughout the manuscript (e.g., amyloid-beta in line 29, whereas in Table 1, it is beta-amyloid).
  4. The legend of Figure 1 and the corresponding main text part should be implemented when discussing the brain region-to-region spreading part.
  5. Figure 2: The authors mention, among the mechanisms underlying protein aggregation in neurodegenerative diseases, that "Molecular chaperones prevent protein aggregation", which sounds a little bit confusing because protein aggregation, as the authors stated, is prevented by proper chaperone function. Conversely,  chaperones with impaired activity contribute to protein aggregation.

Round 2

Reviewer 1 Report

Comments and Suggestions for Authors

Dear authors, thank you for considering my comments and correcting your review accordingly.